# Lessons From A Small-Scale Robot Joining Experiment in VR

Padraig Higgins, Ryan Barron, Don Engel, Cynthia Matuszek

phiggin1,ryanb4,donengel,cmat@umbc.edu

University of Maryland, Baltimore County

Baltimore, Maryland, USA

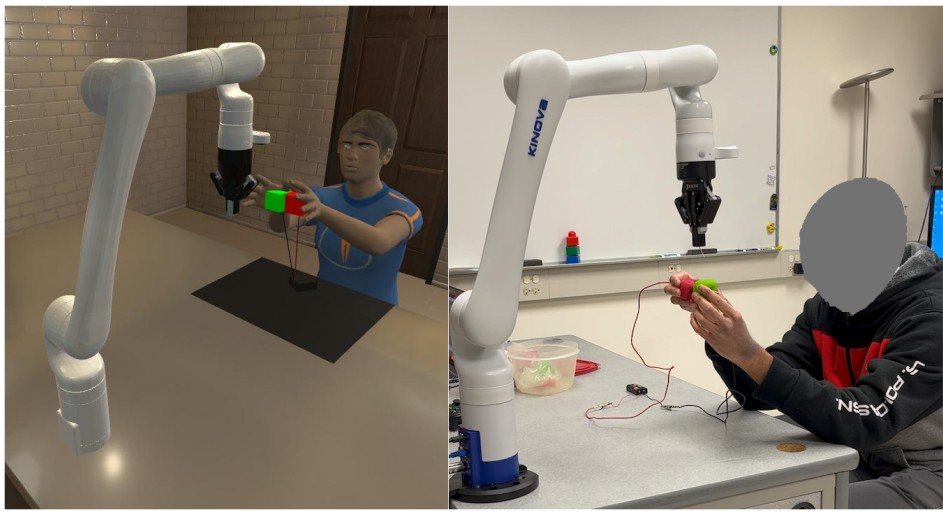

**Figure 1: Participants interacting with the physical and virtual robot to build a simple electric circuit.**

## ABSTRACT

In this paper, we present a shared manipulation task performed both in virtual reality with a simulated robot and in the real world with a physical robot. A collaborative assembly task where the human and robot work together to construct as simple electrical circuit was chosen. While there are platforms available for conducting human robot interactions using virtual reality, there has not been significant work investigating how it can influence human perception of tasks that are typically done in person. We present an overview of the simulation environment used, describe the paired experiment being performed, and finally enumerate a set of design desiderata to be considered when conducting sim2real experiment involving humans in a virtual setting.

## CCS CONCEPTS

• **Computer systems organization** → **External interfaces for robotics**; • **Human-centered computing** → **Virtual reality**.

*VAM-HRI '23, March 2023, Stockholm, Sweden*

© 2018 Association for Computing Machinery.

ACM ISBN 978-x-xxxx-xxxx-x/YY/MM...$15.00

https://doi.org/XXXXXXX.XXXXXXX

## KEYWORDS

Virtual Reality, Sim2Real, Human Robot Collaboration

**ACM Reference Format:**

Padraig Higgins, Ryan Barron, Don Engel, Cynthia Matuszek. 2018. Lessons From A Small-Scale Robot Joining Experiment in VR. In *VAM-HRI '23: International Workshop on Virtual, Augmented, and Mixed-Reality for Human-Robot Interactions, Stockholm, Sweden, 2023.* ACM, New York, NY, USA, 7 pages. https://doi.org/XXXXXXX.XXXXXXX

## 1 INTRODUCTION

Virtual reality (VR) offers a growing opportunity for the development and testing of human-robot interactions (HRI) [Higgins et al. 2021; Mara et al. 2021] as robots become more accessible and affordable in everyday life. The shift of robots from factory to public spaces with the general public on common human tasks requires attention to detail to avoid harm to human collaborators or damage to manipulated objects. However, it is important to be aware of shortcomings that VR-based simulations may have as a reliable surrogate for how humans, robots, manipulated objects, and the environment would behave in the real world. This is why VR-based simulations need to be carefully examined to understand the differences and similarities between VR and real-world problems.

While sim2real has had extensive work [Höfer et al. 2021], the study of human-robot interaction has mainly concentrated on the human experience and perception of the robot during a the non-collaborative interaction [Grzeskowiak et al. 2020; Gérin-Lajoie et al. 2008; Wijnen et al. 2020], leaving the physical or interactional

collaboration between the human and robot more of an open question. To bridge this gap, we conducted a small-scale experiment where a human and robot worked together to manipulate objects, dissimilar to previous work on industrial settings collaboration [Erden and Marić 2011; Lv et al. 2022; Michalos et al. 2018] and larger scale assembly tasks [Raessa et al. 2020; Tsarouchi et al. 2017]. This experiment was carried out in both virtual reality and with a physical robot, and the results were analyzed to determine when behaviors were unique to VR and when they were consistent across platforms.

The experiment was designed to examine the specific scenario of shared manipulation in the context of joining two objects together. The robot acts as an extra hand for the human, which was chosen to be relevant to daily life and focuses on smaller scale physical tasks. Specifically, the human-robot team had to build a simple electrical circuit composed of a battery pack, an LED, wires, and colored conductive putty in red and green, as seen in Figure 2. In this study a robot and human collaborate to construct a simple electric circuit using a battery pack, alligator clips, two molded blocks of colored conductive putty, and a bidirectional light emitting diode (LED) as shown in Figure 2.

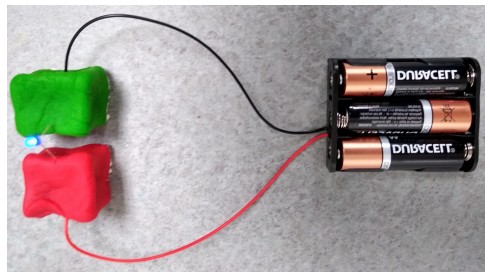

**Figure 2: A simple electrical circuit using a battery pack, a light emitting diode (LED), wires and colored conductive putty. For this task, a person manipulates the putty, and the robot inserts the LED.**

In this paper, we describe the experiment performed and the success and failures of our sim2real transfer. The primary contribution of the work is a set of insights and design guidelines aimed at enhancing the use of VR in studies related to HRI. The study compared the VR and real-world performance of the same small-scale HRI task and revealed a few common modes of failure in simulating HRI tasks in VR.

## 2 RELATED WORK

**Human-Robot Interaction in VR** Simulation has been a useful tool in conducting work in robotics research, reducing the time and expense of acquiring, maintaining, and conducting studies using physical robots. While many of these simulation tools utilize 2D displays [Carpin et al. 2007; Echeverria et al. 2011; Kolve et al. 2017] increasing interest in virtual reality makes it an attractive tool. It can provide greater immersion to users as well as capturing more of the modalities of human interaction than a controller, mouse, or keyboard. The use of 3D tracked hand controllers allows for users to perform various manipulation tasks more intuitively compared

to traditional controllers [Jackson et al. 2018]. This makes virtual reality a better tool to gather demonstrations for learning grasping polices [Stramandinoli et al. 2018; Whitney et al. 2018], as well gathering the training data to learn to perform a sequence of actions [Volmer et al. 2019]. Similar to [Bartneck et al. 2015; Inamura and Mizuchi 2020; Mara et al. 2021; Phan et al. 2018], we utilize the Unity game engine's powerful animation and interaction tools to facilitate the development of complex HRI studies.

**Virtual Reality Compared to the Real World** There has been significant work investigating the differences in how humans behave in real world environments and simulated environments using virtual reality. These have primarily focused on the differences in how humans move about the virtual and real environments [Agethen et al. 2018; Bühler and Lamontagne 2018] or differences in personal space [Gérin-Lajoie et al. 2008]. Previous work comparing human-robot interaction in the real world to virtual reality have investigated the social perception of the real robot versus a virtual one [Wijnen et al. 2020], the differences in proxemics between the real and virtual robot [Li et al. 2019], and the differences in movement in a robot and human following each other in the real world and virtual reality [Grzeskowiak et al. 2020]. In this work we seek to investigate the differences between a collaborative interaction done both in virtual reality and the real world.

There also exist differences between simulation and the real world, the so-called "sim2real gap." While training in simulated environments simplifies and speeds up the process of gathering data, they are not able to fully replicate the nearly infinite variability of the real world. Much of the sim2real work focused on human-robot interaction has focused on training people to use robot systems [Matsas and Vosniakos 2017; Pratticò and Lamberti 2021] or how humans react to robots in different scenarios [Mara et al. 2021; Villani et al. 2018; Weistroffer et al. 2014], which are somewhat less prone to the sim2real gap because humans can conceptualize and generalize from the specific cases present in simulation.

**Collaborative Joining Tasks** While there has been significant work done in collaborative assembly in industrial settings focusing on safety and optimizing workflows. [Peternel et al. 2017] investigates how to optimize for the robot's position during a collaborative task, either in a handover or co-manipulation of a polishing tool. [Erden and Marić 2011] examines a method to handle vibrations of a haptic robot while working with a human to perform an industrial welding task. Work has also been done investigating the overall workflow rather than the act of assembly itself [Timmermann et al. 2021], and the use of human gestures to communicate commands for the robot to alter their execution of the sequential assembly task [Tsarouchi et al. 2017]. Our work differs in that we examine a more granular step of human-robot collaboration, a simple collaborative joining task, and that we are studying the efficacy of VR, relative to real-world teaming, as a tool for exploring this task.

## 3 APPROACH

### 3.1 The RIVR Simulation Environment

In this study we use RIVR [Murnane et al. 2021], a simulator for Robot Interaction in Virtual Reality. RIVR is a VR simulation environment designed to allow a human and robot to interact in

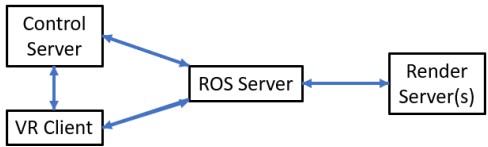

**Figure 3: A diagram showing the major software components of the system and their connections. The control server manages the different simulations; the VR client manages all user input and the scene dynamics; the ROS server launches a rosbridge client and manages interaction with the simulated robot and sensors; and the render servers model the simulated sensor data.**

simulation in a way that is similar to performing the same interactions with a physical robot in the real world. It is designed to allow for studies to be conducted remotely over the internet. To accomplish this, it is built from three primary components, a VR client, render server, and a ROS (Robot Operating System) server as seen in Figure 3. RIVR uses the Unity game engine to both provide the VR interface to the participant, and to generate the visual percepts of the robot, while ROS is used to simulate the robot, and ROS# providing the interface between them.

*3.1.1 System Components.* The ROS server runs an instance of rosbridge to allow for non-ROS clients to connect to the simulator over the internet, as well as managing all the ROS components responsible for controlling the simulated robot, and records timestamped simulation results using rosbag, ROS's built-in data collection tool.

The client is the only piece of software that a participant needs to run locally. It renders the simulated environment to display on the VR headset and captures the audio for the headset microphone. This client is the authority on the state of the scene, running the physics simulation and calculating the position and orientation of any object that is manipulated by either the participant or the robot. This scene state is continuously sent to the ROS server and render server through the rosbridge. The state of the robot, its position and the joint angles for its wheels and arm are all received through the same rosbridge.

Unity provides both a large number of preexisting assets and higher-quality shaders that can better model real-world materials when compared to the typical simulation tools used in robotics, such as Gazebo [Koenig and Howard 2004]. However, having a participant render the simulated environment for themselves, as well as simulating the input of a number of robotic sensors, is expensive in both bandwidth and computation. To get around this, multiple Unity instances can be used as render servers, responsible for rendering both depth and color images from the robot's perspective and sending them over the rosbridge.

*3.1.2 Simulation Components.* As Unity is a freely available game engine, it is designed to be easy to use and to allow quick creation and modification of scenes. The Unity asset store provides a wide range of assets that can be used to build out different environments. It also allows for the ability to modify the scene on the fly, toggling lights on and off, or joining objects together. One of the goals of RIVR is to allow for a robot to interact with a fully embodied

human in the virtual environment. To do this we use MakeHuman,[1] a publicly available tool to build customizable human avatars that can be imported into Unity. Once imported into Unity the avatar was then animated from the poses of the headset and controllers using the Final IK Unity package.[2]

## 3.2 Simulated Experimental Environment

To conduct the study in simulation, the environment and all the necessary components needed to be replicated in a Unity scene. The battery pack, wires, conductive putty, and LED were all modeled using geometric primitives as seen in Figure 4a. Objects that can be manipulated are all "Grabbable." When either the robot closes its hand or the user presses the 'grasp' button on the controller, the distance to all "Grabbable" objects is checked and the closest object within 7.5cm is picked up. This was done by fixing the position of the manipulated object to either the robot's end effector or the one of the hand controllers when picked up, and then disconnecting it when released. To insert the leads from the battery into the putty the user needs to move the lead and putty near each other (within 10cm), at which point the lead is parented to the putty. This only occurs for the appropriate lead and putty, i.e., the lead of the black wire to the green putty or the lead of the red wire to the red putty. Then once joined the lead has the "Grabbable" tag removed. The LED lights up if the tips of the positive and negative leads are each within 3.5cm of the center of the two pieces of putty.

## 4 EXPERIMENT

The first step for the robot involved accurately identifying the positions of the pieces of conductive putty, as they had pre-determined associations with the positive and negative power terminals of the battery pack. This task posed a challenge due to the difficulty in visually distinguishing the LED's terminals. The next challenge for the robot was to correctly position the LED so its leads made contact with the correct putty terminals—a difficult task due to the deformability of the putty, as compared to robot-driven assembly in factories that typically involve connecting rigid objects. The choice to use deformable putty was to reflect real-life small-scale tasks, where many objects are not rigid. In a prior study [Higgins et al. 2023] a standard LED was used, but the small differences between the positive and negative leads caused confusion, so in this study a bidirectional LED was used to avoid confusion.

## 4.1 Experimental Setup

To start the robot initially instructs the participant to pick up the LED and place it between the robots' fingers. It then guides the participant through connecting the battery pack to the conductive putty, placing the positive (red) terminal of the battery pack into the red piece of putty, and placing the negative (black) terminal of the battery pack into the green piece of putty.

Once the battery is connected to the conductive putty the robot prompted the participant to place the two blocks of putty underneath its hand. The robot than attempts to move its hand into a position where it each of the leads of the LED are inserted into the two blocks of putty completing the circuit. Once the robot reaches

[1]http://www.makehumancommunity.org
[2]http://www.root-motion.com/final-ik.html

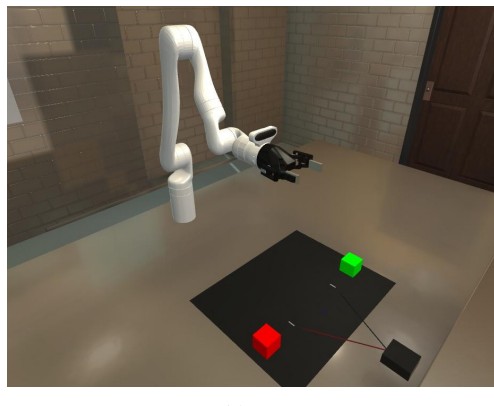

(a) VR

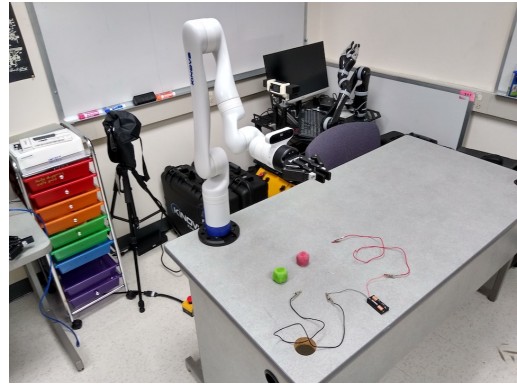

(b) Physical

Figure 4: Experimental setup from the participants point of view, with the robot across a table with the battery pack, wires, red and green conductive putty, and LED between them.

this position it asks the participant if the LED light up. If it was successful it releases the LED, if not it moved back to its initial position and repeat the attempt. Once the task was either completed or the user felt that it was not going to succeed the interaction was finished and a short survey was given.

This task was done both with a simulated robot using virtual reality, and with a physical robot in the real world. Since VR use is not widespread participants using VR were allowed to familiarize themselves with the controls and how to interact with the environment prior to starting the experiment

## 4.2 Experimental Approach

To perform this task a Kinova Gen3 robotic arm with a RGBD camera was used, controlled by the Robot Operating System (ROS). The robot performed all the actions autonomously, but a human controller triggered when each step could start. To determine where to move the arm to insert the LED into the putty, the locations of both pieces of putty were determined using OpenCV color filters to generate a mask for each block of putty. Once generated, the masks were applied to the rectified depth image for the frame, then projected into 3D space forming two pointclouds. From the two pointclouds, the closest two points in the anode and cathode putty blocks were selected and the midpoint between them was used as the target position for the robot arm.

When the LED was ready to be inserted this target position and orientation was used to determine an initial standoff position located 15cm above the target. Once the arm had successfully moved to the calculated standoff position, the manipulator visually servos the end effector to move the LED to the final target position. During both of these actions the robot constantly checks if it can detect both of the pieces of putty and if they are close enough to place the LED; if they are not, the robot verbally asks the user to bring the putty where it can see them or to bring the two pieces closer together.

If the robot was unsuccessful in placing the LED into the putty blocks, the robot could repeat the previous steps until the LED was successfully inserted, at which point the robot released the LED

and moved back to the starting position. The human user held the conductive putty blocks for the duration of the robot's movements. If all the above steps were successful, as soon as the LED leads contacted the anode and cathode putty blocks, the LED lit up.

It is worth noting that, because ROS was developed specifically to control real robots but is incorporated into the RIVR environment for simulating robot actions, the actual code controlling the robot and its behavior is the same regardless of whether the experiment takes place in VR or the physical world. Accordingly, differences in performance between the two settings (discussed next) are not attributable primarily to the robot control code.

## 5 EXPERIMENTAL RESULTS

**Demographics.** Eleven participants were recruited from a common area of a university campus, six female and five male. Ten were aged between 18 and 34, one between 35 and 49. Four identified as white, two as Black or African American, and four as Asian, and one preferred not to say. Five of the participants interacted only with the virtual robot and the other six only interacted with the physical robot.

**Participant Experience Between Settings.** All six who worked with the physical robot were able to complete the task, of the five who worked withe the simulated robot only one was not able to fully complete the tasked. In the VR setting it took an average of $2.6 \pm 2.5$ attempts and $477 \pm 503$ seconds to complete the task, and in the real world it took an average of $2.6 \pm 3.0$ attempts and $206 \pm 136$ seconds.

After the participants finished the experiment they were asked to evaluate how well they understood the instructions given by the robot, how useful they thought the task was, how comfortable they felt interacting with the robot, how frustrating the interaction was, how intuitive they found the interaction, and overall how pleasant they found the interaction using a five-point Likert scale. The results can be seen in Figure 5. In this section we discuss the differences and similarities of the participant experience between the VR setting and the physical world.

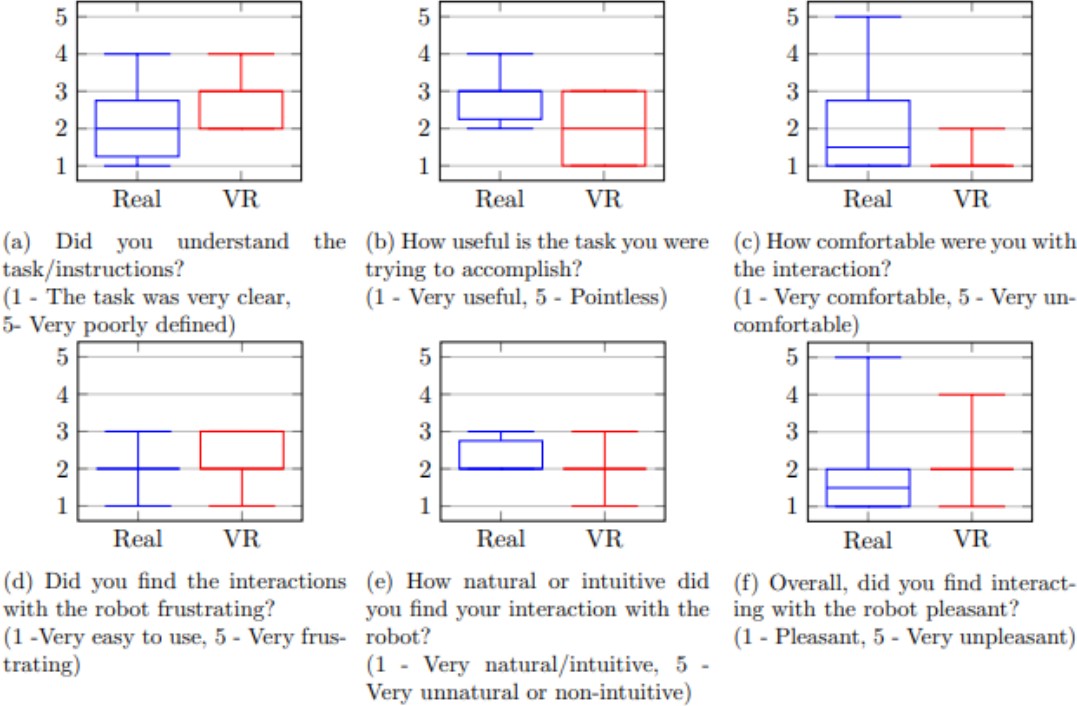

**Figure 5: Post interaction survey responses.**

A one-factor analysis of variance was performed to determine if there were significant differences in responses for virtual and real conditions.

There were not significant differences in how comfortable participants were (p-value of 0.22), understanding of the instructions (p-value of 0.33), how frustrating the interaction was (p-value of 0.66), how intuitive they found the interaction (p-value of 0.39), or how pleasant the interaction was (p-value of 0.81), or in how useful the participants found the task, with participants interacting with the physical robot reporting the task less useful (p-value of 0.14).

The participants who participated in the virtual reality condition were also asked if they had any previous experiences with VR, and of the ten participants, only one had no experience. They were also asked to evaluate the immersiveness and realism of the simulated environment using a five point Likert scale. They reported that while they found the experience fairly immersive (mean 1.8 ± 0.8 1 -I felt like I was there, 5 - Not immersive at all), and realistic (mean 1.4 ± 0.9 1 - Very realistic, 5 - Very unrealistic).

## 6  SIM2REAL CHALLENGES

Throughout the experiments in VR and physical robot tests, several challenges presented themselves. Some of these challenges can be addressed by changing how the virtual environment is engineered, or by adjusting the code controlling the robot in both settings. However, some difficulties are due to fundamental difficulties in bridging the sim2real gap and will be harder to address.

### 6.1  Challenges in Virtual Reality

In a previous pilot study [Higgins et al. 2023] the method used to allow for the user to interact with objects was much simpler, allowing for objects to clip into each other. In this study this system allowed for the collisions of the virtual hands and any objects they are carrying with objects in the environment. While this did not significantly increase the perceived immersiveness, it did increase the perceived realism of the interaction. In both studies users reported the most critical aspect that would increase realism was fully realistic physics.

### 6.2  Bridging the Sim2Real Gap

One of the participants interacting with the real arm noted that at one point it moved unexpectedly toward them, making them uncomfortable and worried that it might poke them with the leads. For other participants, even though they did not explicitly note anything, it was observable that the real arm did not move as smoothly toward the goal position as the simulated arm. The depth images rendered by Unity provide a perfect noiseless image, whereas depth images from physical sensors contain significant amounts of noise, particularly around the edges of objects. In the simulated environment, the size and position of the putty is accurately known due to this lack of noise. In the real world, however, the size and position is less accurate. In some cases, this can result in a miscalculation of where piece of putty is, leading the robot to move non-optimally, and since the task involves close proximity, there is concern the participant might be physically contacted. We suggest that an important element of successfully modeling robotic interactions in

virtual reality is the appropriate rendering of different kinds of sensor noise in the virtual model, as in [Berlier et al. 2022].

It was observed that in the real scenario participants were more likely to ask for clarification about instructions, particularly how to hand over the LED to the robot, while in virtual reality they were more likely to ask for instructions to be repeated. While the participants know that the experimenters are still physically present, in the simulated space they were alone in the scene with only the robot. There may be an even greater difference for studies done with remote participants. Embedding a virtual experimenter in the scene or having the experimenter leave the room can be further investigated.

## 7 LESSONS LEARNED AND FUTURE WORK

The primary purpose of undertaking the study described in this paper was to better understand the role virtual reality can play in research into co-manipulation tasks and other shared physical tasks between a person and a robot. Even with the advent of several VR HRI open source projects, developing a particular VR simulation of an HRI task requires significant time and care. While this overhead should not be underestimated, it will often be substantially less than the cost of acquiring and maintaining a physical robot and a physical test environment. We suggest the following key takeaways:

(1) As in HRI broadly, the importance of performing early, frequent pilot experiments of human subjects studies cannot be overstated. This allows for the mitigation of problems resulting from unexpected human performance and allows early identification of areas where it is appropriate to expend more engineering effort.
(2) Although it is not necessary for every element of a virtual setting to closely mimic the physical reality of a robot's environment, it is important to give early, thorough consideration to questions of where to focus the effort of creating high-fidelity simulations.
(3) During development of virtual environments, it is important to empirically consider a wide variety of physical human behaviors, rather than focusing solely on the behavior that people are expected to display. This will allow the VR setting to be robust to unexpected movements and engagements.
(4) While exploring complex tasks such as shared joining, it is important to consider the built-in limits of the simulation environment (e.g., when addressing soft-body interactions), and to determine whether such interactions are important enough to make modifications to the simulator, or whether to fall back to simpler real-world tasks.
(5) A robot's performance in the physical world is often constrained by the limitations of its sensors, so realistic emulation of noise [Berlier et al. 2022] is important in building simulations where a virtual robot will behave similarly to its real-world analog.
(6) Accurately modeling physics greatly improves the sense of realism, and collisions between manipulated objects should be modeled whenever it is possible.

In future, we intend to pursue extensions of this work intended to provide additional insights and allow application of the lessons learned in this initial work. We will implement a wider variety of tasks on the robot in both reality and VR, both additional single-step tasks and multi-step tasks such as construction of more complex circuits. This will allow us to perform cross-task comparison and understand why some tasks more difficult to perform than others, as well as extending the set of design lessons learned.

One of the complications of VR is the lack of haptic feedback. Some of this could be addressed by using augmented reality for simulation instead of virtual reality. While the robot would still be virtual, a physical table would prevent movement through large-scale static obstacles. Other physical elements of the study, such as the LED or the putty, could be either simulated or physical objects in an AR scene, and the use of physical props versus simulated items could be addressed in a granular way as a further direction of study.

In this work, we explored similarity between VR and physical reality tasks through questions to study participants. A more quantitative approach to measuring similarity would be to have a treatment and control group, where the treatment group does a collaborative task in VR before attempting it in the physical world, and their subsequent performance is compared to a control group who worked only the physical world task without the prior VR experience. Measures could include time to completion of the task and rate of success.

## 8 CONCLUSIONS

We present a small initial study of using virtual reality to study and understand human-robot interactions in a physically collaborative task. This study was performed in similar but non-identical virtual and physical settings, with the goal of understanding how the two contexts are similar and where they differ importantly in systems intended for use in the physical world.

## ACKNOWLEDGMENTS

This material is based in part upon work supported by the National Science Foundation under grant nos. 1940931, 2024878 and 2145642. This material is also based on research that is in part supported by the Army Research Laboratory, Grant No. W911NF2120076.

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
