# OpenReview forum: "Lessons From A Small-Scale Robot Joining Experiment in VR"
_humanrobotinteraction.org/HRI/2023/Workshop/VAM-HRI — VAM-HRI 2023 Oral_

### Official Review · Program_Chairs · 2023-02-25
**Accept**

**Rating:** 8
**Confidence:** 5

**Review:**

Reviewer 1:
This paper examines the differences between a virtual reality simulation of a HRI teamed task and its physical counterpart (constructing a circuit). They report a variety of results and include a list of recommendations/considerations for future studies.


I recommend this paper is accepted. I believe it highlights lessons generally applicable to VAM-HRI studies as a whole which are worth discussing among community members.


Specific comments:
* Please add number of participants in the abstract indicating it is an initial pilot
* “In a prior study [Higgins et al. 2023] a standard LED was used, but the small differences between the positive and negative leads caused confusion, so in this study a bidirectional LED was used to avoid confusion.” - a but of a run on sentence
* I appreciate you putting your exact questions that were asked. Your scales, however, aren’t standardized scales for Likert (e.g., frustrating is not the opposite of useful) and they are technically not scales as each is only 1 item; I would highly recommend reading: https://dl.acm.org/doi/abs/10.1145/3572784
* “In some cases, this can result in a miscalculation of where piece of putty is” -> missing an “a/the” I think
* Please add compensation
* The lessons are greatly appreciated and the most useful part of the paper for other researchers (in my opinion)
* For future studies, I would consider looking at multiple tasks. If you want to claim generalities of tasks for sim2real, it would be much stronger if you had at least 2 different types of tasks in my opinion (this is one of those “what if you changed the robots?” questions that I am not normally a fan of but if the claim is generalize sim2real differences, I would add another task)

------
Reviewer 2:

In this paper, the authors create a small collaborative task between the robot and human, comparing human perception and behavior in both simulation and the real world. They present preliminary findings on the experiment and discuss some challenges to consider for future work that wants to study both simulation and real life experiments.

Overall I think the paper was well written, with an easy to follow outline. The task was also described well! I appreciate the discussion, which should be helpful to other researchers.

One small nitpick is the reference formatting, which doesn't follow ACM numbering guidelines.

---

### Decision · Program_Chairs · 2023-03-02

Accept (Oral)